# Synthesis, Structure and Magnetic Study of a Di-Iron Complex Containing N-N Bridges

**Abhishikta Chatterjee [1], Laurence K. Thompson [2] and Subrata K. Dey [1,*]**

[1]  Department of Chemistry, Sidho-Kanho-Birsha University, Purulia 723 104, WB, India;
    abhishiktachatterjee@yahoo.in
[2]  Department of Chemistry, Memorial University, St. John's, NL A1B 3X7, Canada; lk.thompson@mun.ca
[*]  Correspondence:skdusask@yahoo.ca; Tel.: +91-0947455-7873; Fax: +91-3252-202419

**Abstract:** The iron (II) coordination compound, $\{[Fe(3,6\ pzdc)](H_2O)_2\}_2$. (**1**) has been synthesized from a mixture of $FeCl_2.4H_2O$ and pyridazinedicarboxylate (3,6 pzdc). The molecular structure of complex **1** was determined by single crystal X-ray diffraction study. It reveals that the dinuclear structure contains a pyridazine bridge in between the two metal centers. The variable temperature magnetic study results in g = 2.496(8), J = $-2.50(8)$ cm$^{-1}$, $\theta = -0.1$ K values, by fitting the magnetic data in a simple dinuclear Fe-Fe model which indicates that the major exchange pathway through the N-N bridge. Presence of dense H-bonding interaction leads to supramolecular network formation.

**Keywords:** Iron (II) coordination compound; Molecular structure; Magnetism; Supramolecular structure

## 1. Introduction

The last couple of decades, in coordination chemistry pyridazine has been extensively used as ligands [1–4]. Pyridazine-based and its substituted analogue compounds are explored due to their practical applications in various enzymes regulation processes and have applications in drug design [5–7]. The electron-deficient aromatic systems containing two nitrogen atoms have reduced electron-donor abilities compared to that of pyridines [6]. The two adjacent nitrogen donor atoms in the pyridazine heterocycle attract two metal centers in close proximity [8,9]. With one or two additional donor centers next to the ring nitrogen atoms of pyridazine at 3,6-position leads to tri- or tetra dentate ligands, which have been less explored [10–14]. Symmetric dinuclear complexes of two redox active metal centers have been of interest with regard to the mixed-valence oxidation states [15,16]. For several decades, both experimental and theoretical chemists have been interested in the magnetic properties of dinuclear coordination complexes. A variety of compounds come into sight from the extensive study of different bridging ligands, such as pyrazolates, triazolates and pyrimidine- or pyridazine-linked aimed at fundamental magnetostructural study of iron (II) complexes [17–19]. Transition metal chelates of tetra dentate $N_2O_2$ donor type ligands have been employed as chelating bidentate ligands showed interesting structurally equivalent iron (II) centers with intriguing magnetic properties [20]. A large number of simple di- or poly-nuclear complexes are reported where the metal ions are superexchange coupled through different bridging moiety and most of them exhibit anti-ferromagnetic magneto-structural correlations [7]. Moreover, hydrogen bonding, due to its directional and selective nature is a powerful organizing force in designing useful solid-state materials [21,22]. Construction of a series of supramolecular architecture depends on the choice of spacer (organic moiety) and linker (bridging ligands), and also on the coordination geometries of metal ions [23]. In occurrence of the oxygen atoms in peripheral carboxylate groups, the oxygen atoms might act as potential coordination donors or hydrogen bond receptors. Moreover, to generate the metal-organic coordination networks or molecular building blocks through hydrogen-bonded

supramolecular interactions, the metal-pyridazine moiety plays a very important role as a metallo-ligand [24,25].

Here, we report the preparation of a pyridazine-3, 6-dicarboxylate (Scheme 1) based dinuclear complex of iron with extensive hydrogen boning, leads to an interesting supramolecular network formation, where two rigid dianionic pyridazinedicarboxylate (3,6 pzdc) ligands are organized in a coplanar side-by-side mode around two octahedral metal ions.

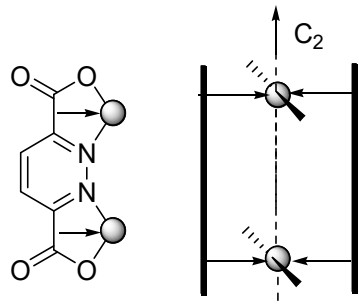

**Scheme 1.** The bis (bidentate) coordination mode of the pyridazinedicarboxylate ligand (3,6 pzdc) and the side-by-side binuclear arrangement.

## 2. Experimental Results and Discussions

### 2.1. X-ray Crystal Structure Description

The key to the complex formation was an exact amount of ferrous chloride tetrahydrate in the mixture of methanol & acetonitrile solution of the pyridazinedicarboxylate (3,6 pzdc). The formation of the desired product was characterized by Elemental analysis (EA), Infrared spectroscopy (IR), Thermo gravimetric analysis (TGA) and Single Crystal X-ray structure determination (SCXRD).

The complex **1** was fully characterized by Single Crystal X-ray analysis. The complex **1** crystallizes in the orthorhombic crystal system with space group Pbca. Structural parameters along with crystal data are listed in Table 1. Single Crystal X-ray structure determination confirmed that during the metalation the ester group of the ligand hydrolyzed to COO- and coordinated as a bis-tetradentate mode to the two iron (II) centers. In this complex both iron(II) centers surrounded by $N_2O_4$ donor sets and bridges by two neutral N1 and N2 atoms of pyridazine moiety in the equatorial plane. The asymmetric unit is shown in Figure 1. The dinuclear complex consists of a center of inversion. Therefore, the two iron (II) centers are crystallographically alike although half of each ligand is unique. The coordination geometry of the dinuclear complex is shown in Figure 2. Both iron (II) atoms are in a distorted octahedral $N_2O_4$ coordination environment, where the two parallel but oppositely aligned ligand strands supplying eight donor atoms and four axial oxygen atoms belong to water molecules. Each Fe (II) ions are member of two five-membered chelate rings of $N_2O_2$ comprising from each of the two ligands. Distorted octahedral ($FeN_2O_4$) coordination sphere is also clear from the values of the bond angles and bond lengths (Table 2). The average equatorial Fe-O bond length is 2.15 Å and axially compressed Fe-O bonds average length is 2.0975 Å. There is no significant difference in the Fe-$N_{pyridazine}$ bond lengths and bonds present at trans to each other having equal bond length. However, the bond lengths of Fe-$O_{equatorial}$ are slightly shorter than those of Fe-$N_{pyridazine}$ bond lengths. Fe-N bond lengths [average 2.195Å] are typical for HS iron (II) ions [26]. The iron atoms in the dimer unit are separated by 3.8947(4) Å. The average equatorial angles fall in the region of 89.99°. The complex possesses coordination environments of the two crystallographically distinct iron (II) sites are very similar due to a center of symmetry located at the mid-point of the complex. The pyridazinedicarboxylate anion has a rich stock of peripheral coordinated and uncoordinated carboxylate oxygen atoms, which are interesting coordination donors and hydrogen bond receptors for the complex which forms hydrogen-bonded supramolecular networks. The complex is highly

stable due to presence of not only strong intramolecular H-bond but also extensive intermolecular H-bonding. Non-bonded oxygen atoms of carboxylate groups are involved in H-bonding interactions with neighboring dimer that is C–O . . . H interactions. The O–H . . . .O interactions are also potent here which gives the complex much more stability. These two types of hydrogen bonding are the main contributing factor for complex 1 in the supramolecular entity formation and stabilization. Hydrogen bonding interactions are revealed in Figure 3 and H-bond distances are listed in Table 3. Hydrogen bonding between these suitable molecules generates multi-dimensional arrays or network. These networks are also connected into a dense 3D network between the axially coordinated water molecule and the carboxylate groups of adjoining sheets. These significantly dense hydrogen bonding interactions lead to supramolecular network formation.

**Table 1.** Crystal Data for the Structure Determination of {[Fe(3,6 pzdc)]($H_2O$)$_2$}$_2$.

| Crystal Data | Complex 1 |
| --- | --- |
| Formula | $C_{12}H_{12}Fe_2N_4O_{12}$ |
| Formula Weight | 515.96 |
| Crystal System | Orthorhombic |
| Space group | Pbca (No. 61) |
| a/Å | 13.458(5) |
| b/Å | 8.721(3) |
| c/Å | 14.760(6) |
| V/Å$^3$ | 1732.3(11) |
| Z | 4 |
| D(calc)/ g cm$^{-3}$ | 1.978 |
| $\mu$(MoK$\alpha$)/mm$^{-1}$ | 1.753 |
| F(000) | 1040 |
| Crystal Size/mm | $0.22 \times 0.18 \times 0.15$ |
| Temperature /K | 293 |
| Radiation/Å | MoK$\alpha$ 0.71073 |
| Theta Min-Max/$^\circ$ | 2.8, 29.2 |
| Data set | −18: 17; −11: 10; −19: 19 |
| Total | 10253 |
| Unique Data | 2072 |
| R(int) | 0.027 |
| Observed[I > 2.0 $\sigma$(I)] | 1821 |
| Nref | 2072 |
| Npar | 136 |
| R1 | 0.0322 |
| wR2 (2sigma < I) | 0.0920 |
| GOF | 1.077 |
| Max. and Av. $\sigma$/esd | 0.00, 0.00 |
| Min. and Max. $\Delta\rho$ [e/Å$^3$] | −0.78, 0.77 |

$w = {}^2(FO^2) + (0.0466P)^2 + 2.2844P]$ WHERE $P=(FO^2 + 2FC^2)/3'$.

**Table 2.** A few Selected Bond Lengths (Å) and Bond Angles (deg) of Complex **1**.

| Bond Lengths/Å | | Bond Angles/deg | |
|---|---|---|---|
| Fe1-O1 | 2.097(1) | O5-Fe1-N1 | 75.54(5) |
| Fe1-O2 | 2.098(1) | O6-Fe1-N2 | 75.58(5) |
| Fe1-O6 | 2.107(1) | N2-Fe1-N1 | 108.54(5) |
| Fe1-N2 | 2.199(1) | O6-Fe1-O5 | 100.33(5) |
| Fe1-O5 | 2.114(1) | O1-Fe1-O6 | 89.47(5) |
| Fe1-N1 | 2.191(1) | O1-Fe1-N2 | 85.21(6) |
| Fe1-Fe1 | 3.8947(4) | O1-Fe1-N1 | 89.10(6) |
| | | O2-Fe1-N2 | 88.66(6) |
| | | O2-Fe1-N1 | 88.70(6) |
| | | O2-Fe1-O5 | 91.93(5) |
| | | O2-Fe1-O6 | 93.25(5) |
| | | O1-Fe1-O5 | 94.50(5) |

**Table 3.** Selected hydrogen bond dimensions in complex **1**.

| D-H···A | D-H/(Å) | H···A/(Å) | D···A/(Å) | <D-H···A/(°) | Symmetry |
|---|---|---|---|---|---|
| O1W-H1W1···O3 | 0.85 | 2.07 | 2.706(2) | 131 | 3/2-x, −1/2+y, z |
| O1W-H2W1···O1 | 0.85 | 2.05 | 2.745(3) | 138 | 1-x, −1/2+y, 1/2-z |
| O2W-H1W2···O1 | 0.85 | 1.99 | 2.794(3) | 157 | x, 1/2-y, 1/2+z |
| C7-H7···O1W | 0.93 | 2.58 | 3.365(3) | 142 | 3/2-x, −y, 1/2+z |

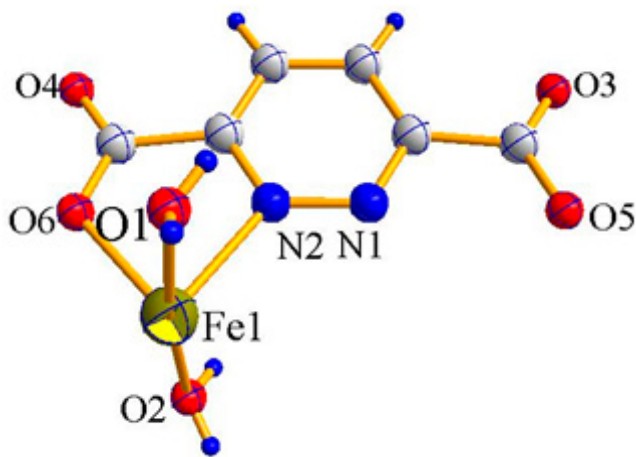

**Figure 1.** Asymmetric unit of complex **1**.

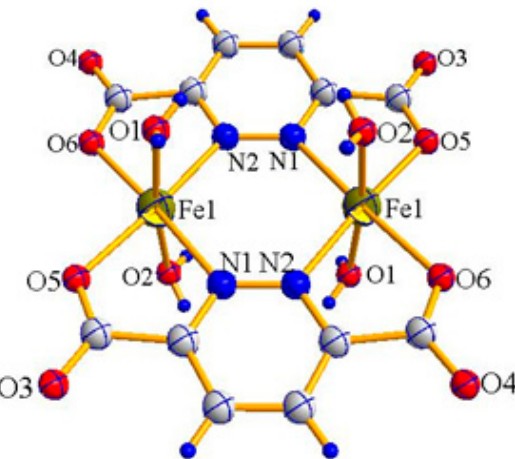

**Figure 2.** Molecular structure of complex **1**.

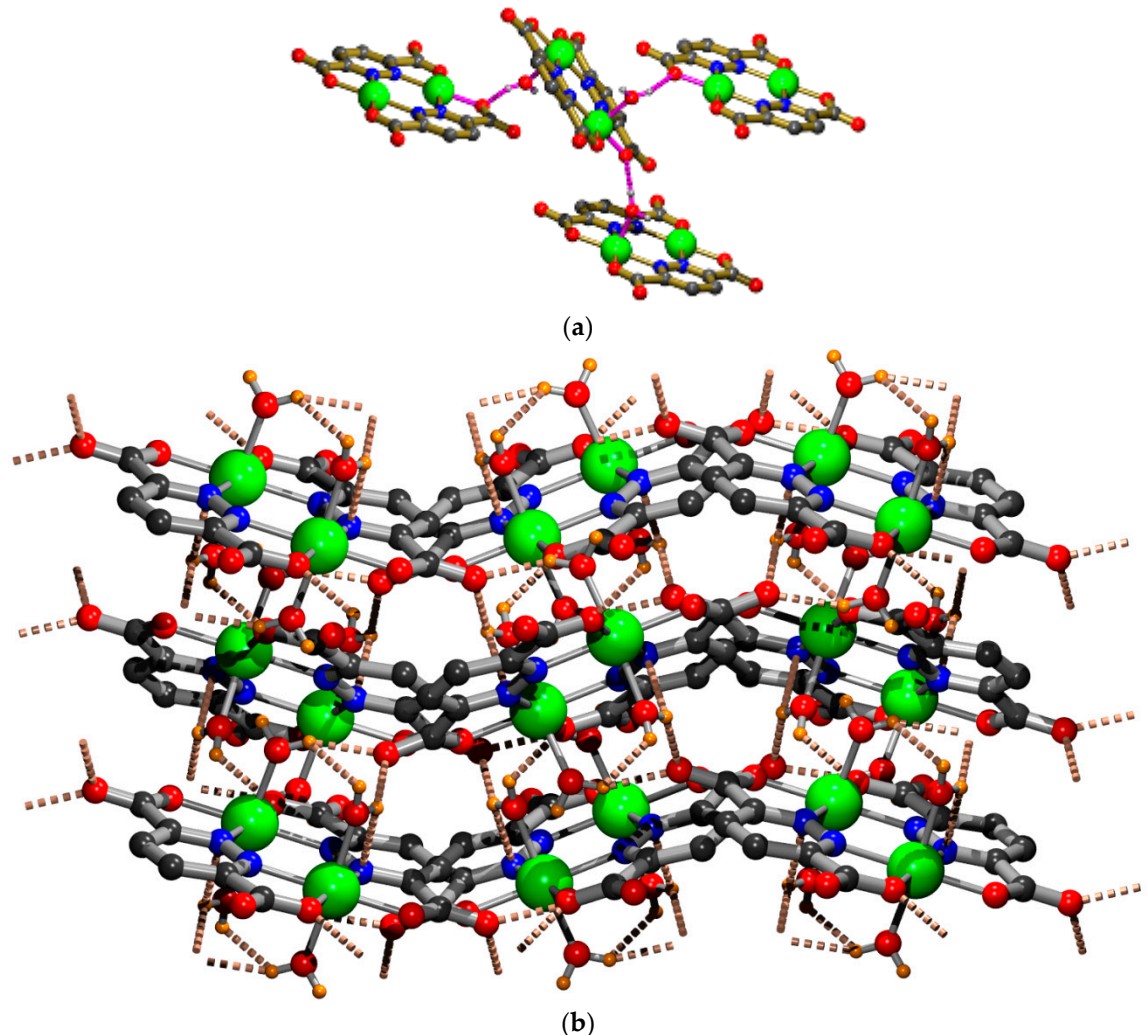

(**a**)

(**b**)

**Figure 3.** (**a**) The coordination environment of the Fe (II) ions in complex **1** showing intramolecular hydrogen bonding interactions. (Strong O-H···O hydrogen bonding interaction). (Blue=Nitrogen, Red=Oxygen, Green=Iron, Yellow= Hydrogen). (**b**) 3D supramolecular structure formed by hydrogen bonding interactions (Strong O-H···O hydrogen bonding interaction). (Blue=Nitrogen, Red=Oxygen, Green=Iron, Yellow= Hydrogen).

## 2.2. IR Spectroscopy Study

We have studied the bonding behaviors of the complex **1** using Fourier transform IR spectroscopy and it reveals all the characteristic bands which were in accordance with single crystal analyses. The appearance of a strong band characteristic for stretching vibrations of C=N at 1638 cm$^{-1}$ instead of the normally observed region (1625–1610 cm$^{-1}$) [27,28] indicates the coordination of pyridazine's nitrogens in complexation. Two peaks in 1638 & 1384 cm$^{-1}$ regions corresponded to the antisymmetric and symmetric COO stretching vibrations, respectively. A peak at 1026 cm$^{-1}$ indicates the presence of pyridazine moiety. Appearance of small intensity band at 833 cm$^{-1}$ indicates Fe-N coordination and band at 550 cm$^{-1}$ due to Fe-O bond. Appearance of a broad band in the region 3435 cm$^{-1}$ is due to stretching vibrations of O-H, which indicates the presence of the water molecule.

## 2.3. Thermal Study

From the Thermo Gravimetric Analysis under non-isothermal conditions it is found that the complex **1** loses four coordinated water molecules in the temperature range 90 °C to 110 °C.

Upon further heating, the final products of decomposition above 380 °C correspond to brown colored metallic oxide of $Fe_2O_3$, which was confirmed by qualitative test.

### 2.4. Magnetic Study

Using the crystalline sample of $Fe_2(pzdc)_2(H_2O)_4$ the variable temperature magnetic susceptibility measurements were performed in the temperature range 2–300 K which is shown in Figure 4 as a plots of molar susceptibility and moment as a function of temperature. As temperature decreases the effective magnetic moment of the complex decreases slowly, representing an antiferromagnetic interaction between the two iron(II) centres. Examining the molecular structure it is clear that the main feature which would lead to such type behavior is the dinuclear $Fe_2(N-N)_2$ subunit, with diazine $N_2$ bridges between the two iron(II) centres. Thus, the magnetic data were fitted to a straight forward model based on a dinuclear exchange Hamiltonian ($H_{ex} = -J \{S_1S_2\}$; $S_1$ = 4/2, $S_2$ = 4/2). The best least squares fit was obtained with J = $-2.50(8)$ cm$^{-1}$, g = 2.496(8), $\rho$ = 0.0001, $\theta$ = $-0.1$ K, TIP = 0, $10^2$ R = 2.19. (where R = $[\Sigma(\chi_{obs}-\chi_{calc})^2/\Sigma\chi_{obs}^2]^{1/2}$; J is the antiferromagnetic coupling constant, $\rho$ = fraction paramagnetic impurity, $\theta$ = Weiss correction). No significant longer-range interactions observed in the extended structure, which might imply additional longer-range exchange. The iron atoms in the dimer unit are separated by 3.8947(4) Å and this large separation rules out any direct metal-metal interaction as the origin of this weak spin coupling. Thus, for the observed antiferromagnetic spin coupling, the most likely pathway is the super-exchange via the N-N bridges.

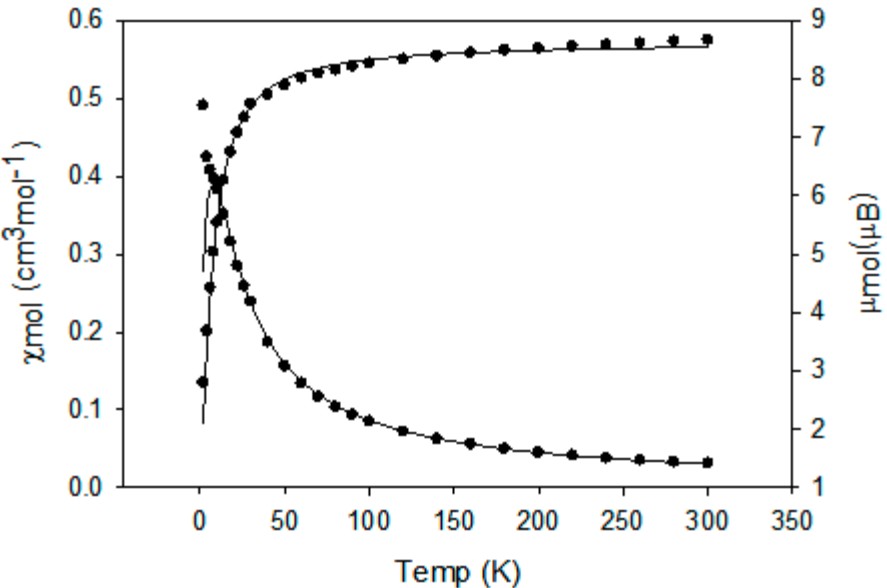

**Figure 4.** Plot of molar susceptibility and moment as a function of temperature.

## 3. Materials and Methods

### 3.1. Materials

Required chemicals and solvents used for the synthesis were purchased from E. Merck, India and used as received without further purification.

### 3.2. Synthesis of {[Fe (L)]. (H_2O)_2}_2

Ligand (3,6 pzdc) was synthesized according to literature procedures [29,30]. Under the dinitrogen atmosphere pyridazinedicarboxylate ligand (3,6 pzdc) (0.748, 2.0 mmol) was dissolved in a hot solution of $FeCl_2.4H_2O$ (0.795 g, 4.0 mmol) in methanol (20 mL) and MeCN (10 mL). Afterwards 2 mmol triethylamine was added to the mixture & dark solution formed. The dark solution was further stirred

with heating for 45 min in dinitrogen atmosphere. Then after cooling the resulting mixture was filtered and allowed for standing at room temperature. After 15 days dark brown crystals were obtained (yield: 48%). Anal. Cald (%) for complex 1: C, 27.93; H, 2.34; N, 10.85; Found: C, 27.98; H, 2.45; N, 10.61.

### 3.3. Single-Crystal X-ray Diffraction Crystallography

X-ray diffraction data of a brown single crystal of complex **1** having dimension $0.22 \times 0.18 \times 0.15$ mm were collected with a "Oxford Xcalibur" diffractometer fitted with graphite monochromatized fine focus Mo Sealed tube Mo-K$\alpha$ radiation ($\lambda_{Mo\,K\alpha} = 0.71073$ Å). Data were collected at 293K temperature using $\omega$ scan technique. Data collection, unit cell refinement and data reduction were performed using CrysAlisPro [31] and CrysAlis RED [32] programs. Using CrysAlis RED [32] multiscan absorption corrections were applied empirically to the intensity values ($T_{max} = 0.775$, $T_{min} = 0.695$). The molecular structure was solved by a direct method and subsequently expanded using the Fourier technique [33]. Refinements of all Non-hydrogen atoms were performed with anisotropic thermal parameters. Water hydrogen atoms were refined with isotropic thermal parameters and geometrically fixed. Molecular graphics were prepared by ORTEP [34], crystallographic illustrations by using CAMERON [35], and publication materials were prepared WinGX [36] software. Further crystallographic data and structure refinement parameter of the complex **1** is summarized in Table 1. A few selected bond distances and bond angles for the complex **1** are given in Table 2.

### 3.4. Magnetic Measurement

Using a Quantum Design MPMS 5 superconducting quantum interference device (SQUID) magnetometer using field strength of 0.1 T variable-temperature magnetic data (2–300° K) were obtained. The crystalline samples were placed in a gel capsule which was secured within a plastic straw attached to the sample rod. Background corrections for the sample holder assembly and diamagnetic component of the complex were applied.

### 3.5. Other Physical Measurements and Calculations

By using a Perkin-Elmer 240 C elemental analyzer elemental analyses (carbon, hydrogen, nitrogen) were performed. FT-IR spectra were also collected by using a Perkin-Elmer FT-IR spectrophotometer in KBr pellets (4000–400 cm$^{-1}$). Thermo gravimetric analyses were carried out with a heating rate of 10 °C/min with a Perkin-Elmer STA-6000 thermal analyser system in a dynamic atmosphere of $N_2$ (flow rate 80 mL min$^{-1}$), the sample was in an alumina crucible, and the temperature range was 25–550 °C. Powder X-ray data was recorded using the Proto AXRD, (Cu K$\alpha$ radiation, $\lambda = 1.5406$ Å) diffractometer and is given in the supplementary file.

### 4. Conclusions

Here, we present the synthesis, crystal structure, low-temperature magnetic study of one symmetrical iron dinuclear complex. From the forgoing discussion, it is found that the N-N from pyridazine moiety coordinated to two iron (II) centres. The dinuclear complex shows moderate antiferromagnetic behaviour and the magnetic interaction is dependent predominantly on the N-N bridges between the two iron (II) centres. It also formed a 3D supramolecular network by extensive hydrogen bonding.

**Supplementary Materials:** For the structural analysis crystallographic data have been deposited with the Cambridge Crystallographic data centre, CCDC No. 1875744. Copy of this information may be obtained free of charge from The Director, 12 Union Road, Cambridge, CB2 1EZ, UK (fax: +44-1223-336033; e-mail: deposit@ccdc.cam.ac.uk or www: http://www.ccdc.cam.ac.uk).

**Author Contributions:** Synthesis and characterizations were done by A.C.. L.K.T. collected and analysed the magnetic data. S.K.D. planned the project and wrote the manuscript.

**Funding:** This research was partly funded by CSIR, New Delhi (Ref. No. 01(2458)/11/EMR dt.16.05.2011) and partly by WB-DST (Memo No. 746(Sanc)/ST/P.S&T/15G dt. 22.11.2016, West Bengal. And the INSPIER fellowship to A.C. (Ref. No. DST/INSPIRE Fellowship/2017/IF170767 dt. 04.07.2018) was funded by DST, New Delhi.

**Acknowledgments:** We are thankful to Rajat Saha, Kazi Najrul University, Asansol for his assistance on X-ray structure. We are also thankful to the Department of Physics, SKBU for providing us the PXRD data. SKD also thanks the Higher Education, West Bengal for funding the Department of Chemistry, Sidho-Kanho-Birsha University.

**Conflicts of Interest:** The authors declare no conflict of interest.

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
