# Peer review of "Synthesis, Structure and Magnetic Study of a Di-Iron Complex Containing N-N Bridges"

_magnetochemistry, doi:10.3390/magnetochemistry4040053_

Round 1

Reviewer 1 Report

In this manuscript, authors describe the synthesis, structure and magnetic property of di-iron complex bridged by pyridazine dicarboxylate. Although several revisions are essentially required in this original manuscript, I judge that this manuscript is suitable for the publication of Magnetchemistry.

1)    There are several structural isomers of pyridazine dicarboxylate. Thus, authors should add the formal name of ligands and its Figure in introduction section. At present manuscript, in the introduction section, readers cannot understand the ligands used in this study.

2)    Synthetic procedure of di-iron complex is briefly explained in the section 3. Authors should describe them in more detail.

3)    Table 1 lacks several information of crystal data of di-iron complex. For example, authors should add the R1 and wR2 (2sigma < I) parameters and GOF parameter.

4)    To confirm the stability and purity of di-iron complex, I recommend the measurement of X-ray powder diffraction (XRD) of di-iron complex. Of course, authors should make a comparison between obtained XRD data and simulated data of single crystal X-ray diffraction analysis. This result is very important information because reader cannot understand the crystalline quality of sample used in the magnetic measurement. 

Author Response

Answers to the Reviewers’ questions: Manuscript ID: magnetochemistry-390295 Type of manuscript: Article Title: Synthesis, structure and magnetic study of a di-iron complex containing N-N bridges Thanks a lot to both the Reviewers for their constructive comments. We have gone through the comments and revised the manuscript accordingly. We have improved the introduction part and modified the result section also along with the addition of related references. We have also checked the grammatical part. Here I am giving a brief outlines about the changes: Reviewer 1: In this manuscript, authors describe the synthesis, structure and magnetic property of di-iron complex bridged by pyridazine dicarboxylate. Although several revisions are essentially required in this original manuscript, I judge that this manuscript is suitable for the publication of Magnetchemistry. There are several structural isomers of pyridazine dicarboxylate. Thus, authors should add the formal name of ligands and its Figure in introduction section. At present manuscript, in the introduction section, readers cannot understand the ligands used in this study. We have added a formal name along with its figure in the introduction section (Scheme-1). Synthetic procedure of di-iron complex is briefly explained in the section 3. Authors should describe them in more detail. Synthetic procedure of the complex is rectified. Thanks for the input. Table 1 lacks several information of crystal data of di-iron complex. For example, authors should add the R1 and wR2 (2sigma < I) parameters and GOF parameter. All data are given in the Table 1. To confirm the stability and purity of di-iron complex, I recommend the measurement of X-ray powder diffraction (XRD) of di-iron complex. Of course, authors should make a comparison between obtained XRD data and simulated data of single crystal X-ray diffraction analysis. This result is very important information because reader cannot understand the crystalline quality of sample used in the magnetic measurement. Nature of the sample is crystalline and we did all the experiments from the same batch of the sample. So we did not run the PXRD data initially. Anyway, according to the suggestion I have attached the PXRD data in the supplementary data file.

Reviewer 2 Report

The paper describes the structure and magnetism of dinuclear iron(II) pyridazine bridged complex with terminal carboxylate goups. The authors must confirm that they used iron(III) nitrate as starter (p. 6, line 140), and must make comment in regard to what is the reducing agent.

The introduction should include citation to the dinuclear iron(II) work of Schneider et al (Eur J Inorg Chem 2013, 850-864).

The English needs checking e.g. p. 1 line 23  The last couple of...; p. 5 line 119   ..were performed

There is just enough new material to make the paper acceptable to Magnetochemistry

Author Response

Answers to the Reviewers’ questions: Manuscript ID: magnetochemistry-390295 Type of manuscript: Article Title: Synthesis, structure and magnetic study of a di-iron complex containing N-N bridges Thanks a lot to both the Reviewers for their constructive comments. We have gone through the comments and revised the manuscript accordingly. We have improved the introduction part and modified the result section also along with the addition of related references. We have also checked the grammatical part. Here I am giving a brief outlines about the changes: Reviewer 2: The paper describes the structure and magnetism of dinuclear iron(II) pyridazine bridged complex with terminal carboxylate goups. The authors must confirm that they used iron(III) nitrate as starter (p. 6, line 140), and must make comment in regard to what is the reducing agent. We are grateful to the reviewer for pointing out the mistake. It was completely overlooked and extremely sorry for that. However, we have re-written the synthesis part. The introduction should include citation to the dinuclear iron(II) work of Schneider et al (Eur J Inorg Chem 2013, 850-864). Mentioned reference along with some other suitable references have been added. The English needs checking e.g. p. 1 line 23 The last couple of...; p. 5 line 119 ..were performed We have checked the manuscript and modifiedaccordingly. There is just enough new material to make the paper acceptable to Magnetochemistry

Round 2

Reviewer 1 Report

Suitable revisions for publication were confirmed in the revised manuscript. However, Scheme 1 is not included in this manuscript. So, authors should add the Scheme 1 in the introduction part.

Author Response

Thanks to the Reviewer for the acceptance and for the comments. Comments and Suggestions for Authors Suitable revisions for publication were confirmed in the revised manuscript. However, Scheme 1 is not included in this manuscript. So, authors should add the Scheme 1 in the introduction part. Ans: I guess the figure of the scheme is missing from the revised version where we had included the Scheme 1. Anyway here we have again included the Scheme 1 in the 2nd revised version accordingly.